# The Concept of «Peritumoral Zone» in Diffuse Low-Grade Gliomas: Oncological and Functional Implications for a Connectome-Guided Therapeutic Attitude

**DOI:** 10.3390/brainsci12040504

**Published:** 2022-04-15

**Authors:** Melissa Silva, Catalina Vivancos, Hugues Duffau

**Affiliations:** 1Department of Neurosurgery, Hospital Garcia de Orta, 2805-267 Almada, Portugal; smelissa.msilva@gmail.com; 2Department of Neurosurgery, Hospital Universitario La Paz, 28046 Madrid, Spain; catalina.vivancos@gmail.com; 3Department of Neurosurgery, Gui de Chauliac Hospital, Montpellier University Medical Center, 34295 Montpellier, France; 4Team “Plasticity of Central Nervous System, Stem Cells and Glial Tumors”, Institute of Functional Genomics, National Institute for Health and Medical Research (INSERM) U1191, University of Montpellier, 34295 Montpellier, France

**Keywords:** low-grade glioma, peritumoral zone, surgery, brain connectome, epilepsy, quality of life

## Abstract

Diffuse low-grade gliomas (DLGGs) are heterogeneous and poorly circumscribed neoplasms with isolated tumor cells that extend beyond the margins of the lesion depicted on MRI. Efforts to demarcate the glioma core from the surrounding healthy brain led us to define an intermediate region, the so-called peritumoral zone (PTZ). Although most studies about PTZ have been conducted on high-grade gliomas, the purpose here is to review the cellular, metabolic, and radiological characteristics of PTZ in the specific context of DLGG. A better delineation of PTZ, in which glioma cells and neural tissue strongly interact, may open new therapeutic avenues to optimize both functional and oncological results. First, a connectome-based “supratotal” surgical resection (i.e., with the removal of PTZ in addition to the tumor core) resulted in prolonged survival by limiting the risk of malignant transformation, while improving the quality of life, thanks to a better control of seizures. Second, the timing and order of (neo)adjuvant medical treatments can be modulated according to the pattern of peritumoral infiltration. Third, the development of new drugs specifically targeting the PTZ could be considered from an oncological (such as immunotherapy) and epileptological perspective. Further multimodal investigations of PTZ are needed to maximize long-term outcomes in DLGG patients.

## 1. Introduction

As the name states, diffuse low-grade gliomas (DLGG)—World Health Organization (WHO) grade 2 diffuse astrocytic and oligodendroglial tumors [1]—are in essence poorly circumscribed and intrinsically heterogeneous [2,3,4,5] tumors that progressively infiltrate the brain. Regardless of the common initial slow growth rate, these tumors will ineluctably become more aggressive with malignant transformation if left untreated [6].

Concerning their typical features on routine magnetic resonance imaging (MRI), they may even seem somewhat delineated T1-hypointense and T2- and FLAIR-hyperintense lesions, but in fact, this imaging technique only identifies the high-density tumor, underestimating the real spatial extent of the disease [7]. Indeed, it has clearly been shown that isolated tumor cells extend beyond the margins of the tumor depicted on MRI [8,9,10,11,12,13], leading to the concept of so-called peritumoral zone (PTZ) at the interface between the tumor core and the surrounding healthy brain.

Current studies are attempting to find physiological/metabolic imaging modalities that can help overcome this MRI limitation and thus correctly identify tumor limits [4,14,15,16], which would eventually lead to more accurate assessments on therapeutic goals and subsequently improve its results.

In fact, it has been evidenced that the extent of resection is a major prognostic factor, and hence, surgery is highly recommended in DLGG [17]. Furthermore, recent reports state that, when feasible, a “supratotal” resection (SpTR)—meaning a functional-guided resection beyond MRI signal abnormalities and therefore including the PTZ—may have an even greater impact on the natural history of the disease [12,18,19,20,21,22]. In this case scenario, it is also likely that adjuvant treatment, namely, chemo- and/or radiotherapy, could be postponed and thus further extend not only survival but, moreover, patients’ long-term quality of life (QoL) [23].

In addition, there has been increasing evidence that the associated epileptogenesis in DLGG is highly related to the increased and aberrant excitatory synapses observed in the peritumoral area [24,25,26,27,28,29], which makes it another therapeutic challenge, especially by supporting SpTR also for functional reasons since seizure control is strongly related to QoL.

The aim of this review is therefore to assemble the current knowledge regarding the PTZ in DLGG and to investigate the oncological and functional implications of this concept in multimodal therapeutic management.

## 2. DLGG Heterogeneity and Relation with the Surrounding Brain: The Peritumoral Zone (PTZ)

DLGGs (WHO grade 2 gliomas) are infiltrative tumors arising from glial cells of the brain that are traditionally classified according to their cell morphology—astrocytes, oligodendrocytes, or a combination of both. More recently, their categorization includes molecular features, especially IDH mutation and 1p19q codeletion status [1]. Indeed, the 2021 fifth edition of the WHO classification introduced major modifications that advance the impact of molecular diagnostics in brain tumor, it established different approaches to glioma nomenclature and grading, and it emphasized the importance of integrated diagnoses: new tumor types and subtypes were introduced based upon new diagnostic technologies, such as DNA methylome profiling [1].

Regarding glioma spatial structure, defined by its growth pattern, three main types are considered: solid tumor with no peripheral isolated tumor cells (ITCs), tumor tissue with peripheral ITCs, and ITCs within intact brain parenchyma (meaning no solid tumor) [30]. When analyzing the behavior of tumor cells’ growth according to histology, it was found that oligodendroglial tumors with 1p19q loss tend to be more circumscribed lesions with a predominant proliferation in situ, in contrast to the more diffuse, infiltrative, and less bulky astrocytic ones [31]. In a study of histologically defined glial tumor growth patterns, those with mixed or “solid” growth patterns were also more likely to have 1p/19q loss, and those with infiltrative growth were more likely to have intact 1p/19q [32]. A more recent work demonstrated that DLGGs with sharp borders were more frequently IDH-mutant compared with tumors with indistinct borders [33].

When studying the intratumoral heterogeneity of DLGGs, within a WHO grade 2 tumor, there can be single or multiple microfoci of higher cellular or vascular density or even atypia (the presence of which was related to significantly lower survival) [3]. In a series of low-grade oligodendroglioma biopsies, it was found that in 62.5% of cases, the cycling cells formed a ring of proliferation in the peripheral areas of the tumor, whereas in the remaining cases the cycling tumor cell fraction increased toward the center of the tumor core, evidencing the heterogeneity of these tumors [2]. Additionally, concerning oligodendrogliomas, but using en bloc resected tumors, different tumor cell and vessel densities were identified throughout all the cases [4]. A more recent study revealed the existence of sparse, but widely distributed, protoporphyrin IX “hotspots” within low-grade gliomas, which exhibited some malignancy features, such as less differentiated cellular state, ability to divide, and metabolic reprogramming [5].

With the knowledge of isolated tumor cells within intact brain parenchyma, the concept of “peritumoral zone”, mostly applied for glioblastomas [34], can also be used for DLGG to describe a peripheral area with the same macroscopic aspect of a normal brain but already with microscopic tumor infiltration. This is usually a radiological definition.

In fact, it is nowadays understood that DLGGs actually extend beyond MRI-defined abnormalities. In 1987, in a series of stereotaxic biopsies from glial neoplasms—which included 20 grade 2 lesions—the authors took 5 samples from normal T2-weighted MRI regions surrounding grade 2 astrocytomas and in 3 of those cases found ITCs, admitting that “normal” brain tissue was rarely biopsied for ethical reasons [8]. Five years later, Watanabe et al., saw tumor cells already infiltrating brain tissue beyond the T2 high-intensity lesions delineated on MRI in 5 of their 8 low-grade-glioma cases [9]. When trying to determine the helpfulness of diffusion tensor imaging (DTI) in better identifying glioma margins, Price et al. (2006) found tumoral cells in 8 of the 18 patients in whom the biopsy trajectories were taken into peritumoral areas (with normal T2 signal intensity on MRI), but in their series, this was less seen in lower grades—1 of 7 cases [10]. In 2010, in a study of diffuse low-grade oligodendrogliomas, Pallud et al. (2010) collected biopsy samples from within and beyond the hypersignal areas on T2-weighted and FLAIR MRI sequences and found the presence of cycling cells beyond MRI-defined abnormalities in all patients—where a higher density of tumor cells was found at distances of 10 to 20 mm beyond the imaging limits but not at distances greater than 20 mm [11]. Similarly, Gerin et al. (2013) discovered cycling tumor cells infiltrating the parenchyma around the tumor core up to 20 mm outside the MRI-defined abnormalities [2]. In 2016, Zetterling et al., also found that tumor cells extended beyond the FLAIR MRI border in all the 5 cases of en bloc resection studied at a maximum distance of 1.4 cm from the radiological margin. Additionally, a common pattern of growth was noted, as the tumor cells followed the white matter (WM) tracts and were slightly more concentrated in the peripheral parts of the tracts [13]. Importantly, in a recent series using intraoperative image-guided biopsies, genetic analyses using RNA sequencing and whole-exome sequencing observed a gene expression pattern and mutational landscape of the PTZ that were distinct from that seen in the tumor core and peripheral brain tissue [15].

Studies on PTZ patterns of WM displacement and/or invasion may be important for understanding how much functional brain tissue is compromised, because invasion of the WM tract connectivity prior to surgery represents a main limitation of neuroplasticity [35]. Latini et al. (2021), when trying to understand whether specific features of the different WM pathways could reflect the differences in observed glioma infiltration, built a theory that a smaller fiber diameter, decreased fiber density and increased extracellular space may represent pathways of least resistance for glioma cell dissemination [36]. In the same spirit, a recent review on DLGG interaction with WM tracts suggested that myelin may constitute a protection against glioma cell migration; therefore, its nonexistence or destruction could result in fragility sites facilitating tumor invasiveness [37].

Regardless of the common initial slow growth rate, DLGG will finally become more aggressive with malignant transformation if left untreated [6]. That is the main reason why it is important to thoroughly understand intratumoral and peripheral cellular behavior and its complex interactions with neural networks. 

## 3. FLAIR MRI Cannot Reflect the PTZ: How Can DLGG Delineation Be Improved?

Following the awareness of the conventional MRI’s inability to delineate DLGG’s real margins, several groups have explored physiologic and metabolic imaging modalities to surpass this limitation. 

Radiomics, which aims at extracting multiple quantitative imaging features using reproducible algorithms, has been increasingly applied and represents the basis of radiogenomics, whose purpose is to determine the association between the collected imaging data and both genomic signatures and molecular phenotypes of gliomas [38,39]. Of note, there are emerging methods based upon a deep learning and radiomic model that could be promising for glioma grading using multiplanar reconstructed MR contrast-enhanced T1-weighted imaging [40].

For each unique MRI sequence, different maps can be generated, and contemporary studies are trying to figure out which one(s) is(are) most useful on a glioma study. Some well-known examples are apparent diffusion coefficient (ADC) maps from diffusion-weighted imaging (DWI); maps of fractional anisotropy (FA), mean diffusivity (MD), and kurtosis (K) from DTI; and regional cerebral blood volume (rCBV) maps from dynamic susceptibility contrast-enhanced (DSC) sequences.

Within the range of MRI sequences, DWI and DTI have been the most frequently used sequences to improve the detection of tumoral infiltration in both low- and high-grade gliomas (HGGs). In fact, DWI was included in the 2015 consensus recommendations as part of the minimum standard brain tumor imaging protocol [41]. ADC and MD values are considered indirect measures of tumor cellular density because proliferating tumor cells hamper the diffusion of extracellular water. Therefore, they estimate tumor proliferation in DLGG by inverse correlation. Of note, brain edema is an important confounder. Throughout the years, studies on this matter have been contradictory. A recent systematic review and meta-analysis concluded that ADC derived from DWI has a high diagnostic performance in differentiating low-grade from high-grade gliomas, underlining its easy accessibility and lower cost when compared with other metabolic and physiologic imaging [42]. However, the diagnostic accuracy regarding PTZ was not scrutinized, possibly because of the scarce literature focusing on this specific subject. Interestingly, a study on tumor proliferation that focused exclusively on DLGG found that the interval changes of ADC values correctly predicted disease progression (diagnostic accuracy of 86%) before apparent radiologic progression on conventional imaging [43], supporting the potential use of radiomics in detecting peritumoral cells.

Another explored technique is diffusion kurtosis imaging (DKI), an extension of DTI, which provides quantitative data on how tissue water diffusion deviates from a normally distributed diffusion. A prospective study published in 2017, which compared DKI parameters with perilesional normal-appearing white matter (NAWM) and contralesional NAWM, found some significant differences, namely, higher mean diffusivity and lower kurtosis in the perilesional WM, which were associated with tumor infiltration [14]. Some of the latest studies combining DWI and DKI parameters, despite including DLGG in their cohorts and evaluating the peritumoral area, essentially reinforced the ability of the techniques in differentiating DLGG from HGG but were unable to draw new conclusions regarding the PTZ in DLGG [44,45].

The DSC sequence, that is, a perfusion-weighted imaging (PWI) technique, identifies changes associated to neoangiogenesis, which correlate with malignancy. The resultant quantitative parameters (including rCBV) are potentially helpful in predicting grading, progression, and prognosis in DLGG, as documented in a 2015 systematic review [46]. A prospective pilot study with 10 subjects (4 with DLGG and 6 with HGG) used multimodal MRI to build a predictive model of tumor infiltration in glioma patients, including DTI and PWI quantitative metrics, that could serve as a biomarker for nuclear density [47]. Another prospective work, from 2018, used multiparametric MRI from 7 patients (5 with DLGG and 2 with HGG) to assess each sequence’s capacity to differentiate between tumor core, tumor infiltrated edema, and normal tissue. In this series, the only statistically significant MRI-derived feature able to differentiate tumor core from infiltrated edema was rCBV, with a specificity of 95%; however, it was not able to perform likewise in discerning tumor infiltrated edema from normal tissue [48].

Of note, using multicomponent T2-weighted MRI relaxometry (a method of myelin water imaging) in proton therapy, Bontempi et al., accidentally found that decomposing T2 can be more sensitive than conventional FLAIR imaging for detecting subtle tissue alterations in the peritumoral region of WHO grade 2 and 3 gliomas [16].

In addition to all these advanced MRI sequences, functional MRI (fMRI) and DTI tractographies are nowadays part of the presurgical imaging of DLGG in many institutions. Resting-state fMRI and advanced high angular resolution diffusion imaging (HARDI) tractography are refining these methods’ results. They are useful in investigating cortical and subcortical plasticity and exploring the association between longitudinal functional changes and progression of disease [35,49,50]. One group recently reported the use of mean blood-oxygen-level-dependent (BOLD) signal from resting-state fMRI (rs-fMRI), coupled with conventional structural MRI, to compare not only with histology results but also with RNA sequencing and whole-exome sequencing of biopsy samples from tumoral (TT), peritumoral (PT), and healthy (HT) tissues. The authors found that the mean BOLD signal was significantly associated with the molecular data (calculated expression similarity index of differentially expressed genes) in their determination of PT relative to HT and TT (*p* < 0.001). In their series, the average PT distance in DLGG was 10.9 mm with an 18% recurrence rate, whereas in HGG it was 8.75 mm with an 81% recurrence rate at the last follow-up. The Kaplan–Meier curves showed a significant difference in progression-free survival between patients with a PT distance ≤10 mm and >10 mm [15].

Moreover, positron emission tomography (PET) imaging, a noninvasive method of measuring biochemically specific targets of tumors (and more specifically PET imaging with labeled amino acid tracers), has been widely used to depict the biological activity of DLGG [4,51,52]. In a recent prospective study, Verberg et al., sought to determine, through multiregional biopsies, the most accurate imaging combination to detect glioma infiltration. In their series, fluoroethyl-L-tyrosine PET was not found to be a component of the optimal imaging combinations for nonenhancing glioma, and its diagnostic accuracy was lower than that of FLAIR MRI [53]. To our knowledge, no other relevant studies have been published on the value of PET imaging in identifying the PTZ on DLGG. Additionally, the present limited availability of amino acid PET makes it a nonroutine imaging technique on glioma management. 

Closing this topic, we highlight a 2021 systematic review on the integration of multimodality imaging and artificial intelligence to improve the visualization of tumor cell infiltration in glioma [54], which found only two articles that included DLGG (along with HGG) in their cohorts [47,53], depicting the gap between low-grade and high-grade research.

## 4. PTZ Recognition Supports Supratotal Resection and Vice Versa

Even if it is not possible to identify the full extent of the PTZ on preoperative scans, the existence of peripheral ITCs cannot be denied. Therefore, when it comes to treating this disease and knowing that, when feasible, surgery is the first line of treatment [17], the concept of total resection in DLGG is somewhat misleading. Indeed, recent proposed recommendations on the subject maintain that the definition of complete resection of WHO grade 2 and 3 gliomas requires complete removal of the T2/FLAIR-hyperintense MRI lesion [55]. Understandably, as long as MRI is the best available imaging technique for evaluating DLGG, this definition is the most acceptable one, even if in reality this degree of extent of resection (EOR) leaves several ITCs behind.

In spite of the latest Cochrane review (2017) on biopsy versus resection for low-grade gliomas, stating that there are no randomized controlled trials to determine the best surgical choice [56], numerous systematic reviews and meta-analysis confirm a positive correlation between EOR and overall survival (OS) in DLGG [22,57,58,59], along with some of the most recently published single-institution experiences [60,61,62,63,64]. The longer OS achieved is related to a change in the natural history of the disease, surgery allowing us to significantly reduce the risk of malignant transformation, thanks to a decrease in the number of tumor cells. Additionally, taking into account PTZ knowledge, earlier treatments should be considered before a too wide extension of this tumor cell outskirts, making maximal resection impossible [20].

Therefore, bearing in mind the previously cited works describing a PTZ margin of at least 1–2 cm in DLGG, a surgical resection extending beyond the MRI abnormality margin would lead to an even larger number of tumoral cells removed and a greater benefit on OS. Considering this, several teams, listed as follows, have been pursuing a “supratotal” resection (SpTR), when functionally feasible, resulting in better oncological outcomes. First, Yordanova et al., compared the oncological outcomes of 15 patients following a SpTR of a DLGG with those of a control group of 29 patients with gross total resection (GTR) and found a statistically significant reduction of anaplastic transformation and of the need for adjuvant treatment in the SpTR group (mean postoperative follow-up of 36 months) [12]. From the same institution, in 2006, the long-term outcome results of a series of DLGG patients after SpTR was retrospectively analyzed. Half of the cases had no tumor relapse, and of the ones that recurred, none displayed malignant transformation [18]. In these two reports, there was no permanent neurological worsening, and all patients returned to normal life. In a confirmatory study published in 2020, Rossi et al., found that a SpTR was achieved in 32.3% of 449 presumed DLGGs (79.5% actual DLGGs) with a rate of 0.68% permanent neurological deficits, and assured that SpTR is possible in a considerable number of patients when a functional approach is used [19]. On the following year, a study from the same institution, encompassing 319 IDH-mutated presumed low-grade gliomas (mean follow-up of 6.8 years), observed that 82.4% of patients progressed on the partial/subtotal resection group in contrast with 5.4% on the SpTR group [21]. These reports on SpTR share an understanding on the feasibility and safety of the procedure since they all rely on functional-guided surgery using direct electric stimulation (DES) during awake procedures. Indeed, it has extensively been demonstrated that intraoperative DES represents a unique opportunity to directly investigate cerebral functional connectivity [65], resulting in the proposal of new models of brain connectome based upon the dyssynchronization within and between neural networks elicited by DES [66]. In clinical practice, this novel concept led to a paradigmatic shift, that is, to achieve a surgical resection up to the identification of the neural circuits critical for brain processing identified by DES [67], and not according to the data provided by neuroimaging, which underestimates glioma infiltration. In other words, a maximal safe surgical resection means «connectome-based surgery» (and not «DLGG core-based surgery»), with the aim of removing the PTZ if peripheral ITCs did not yet invade functionally critical cortico-subcortical neural networks [68] (Figure 1). Therefore, the onco-functional balance is at its highest level when it comes to SpTR for DLGG, under such conditions, as it translates into the largest possible supramarginal removal while preserving functional boundaries [69].

Indeed, a 2021 review on brain mapping-aided SpTR of brain gliomas (including DLGG and HGG) highlighted a longer OS, thanks to maximal surgery and its improved safety because of intraoperative neuromonitoring [20]. This is comprehensibly truer for DLGG, as its slow tumor growth rate induces functional reorganization of subcortical networks and thus higher tolerance of SpTR [20]. The surgical removal of the PTZ becomes even more important when considering the already-mentioned study by Gerin et al., which observed in almost two-thirds of their oligodendroglioma cases that the cycling tumor cell fraction was higher at the limit of the MRI-defined abnormalities [2]. Regarding the known patterns of invasion of DLGG, SpTR will, possibly, be more successful in controlling the disease on those cases where the proliferative pattern is dominant, in opposition to more infiltrative ones [69].

To sum up, the ultimate aim of SpTR is to have a bigger impact on the natural history of DLGG and postpone malignant transformation for as long as possible while preserving the QoL.

## 5. PTZ Influence on (Neo)Adjuvant Treatment

From an oncological point of view, if surgery is the first line of treatment and SpTR is successfully achieved, adjuvant treatments, such as chemo- or radiotherapy, could be postponed. Indeed, in a recent European survey on DLGG, 80% of responders affirmed that in case of a total resection of the FLAIR lesion, they would recommend a “wait-and-watch” strategy afterwards [70]. This percentage would probably be higher when considering SpTR.

As regards chemotherapy (CT) regimens, two main modalities are well accepted and nowadays widely used in DLGG: mono-chemotherapy with temozolomide (TMZ) and the association of procarbazine, lomustine, and vincristine (PCV) [70]. A 2020 literature review on the use of CT alone found lower rates of disease progression under TMZ (4%–5%) when compared with PCV (11%–18%) [71]. Nonetheless, there is no consensus when choosing a reference treatment [70,72]. Regarding radiotherapy (RT), despite its impact on glioma, there is growing evidence that it may increase the risk of neurocognitive side effects in the long term [73]. A review on the subject found that one of the major mechanisms of radiation-induced brain injury is WM degeneration in and beyond the PTZ, which impairs the neuroplastic potential—due to a lower potential of functional reorganization at the level of the subcortical WM pathways compared with the cortical level [74]. Despite technical advances in RT to improve neurocognition preservation, there is still little regard for the brain connectome when planning the irradiation sites [74], as opposed to the surgical management of DLGG that is nowadays guided by functional boundaries [68,69]. Furthermore, in terms of survival, there is no clear benefit on early versus late RT in DLGG, as detailed in the latest Cochrane review [75]. It is believed that there might be a synergistic effect when using chemotherapy with concomitant radiation therapy, but until now, there are no prospective studies formally comparing chemotherapy alone with chemoradiation in DLGG. Some authors have been considering results from the radiation therapy oncology group (RTOG) 9802 trial for “high-risk” patients as a justification for the use of chemoradiation on this subgroup of patients [76]. However, the definition of “high-risk” DLGG remains arguable [77].

All things considered, if PTZ is better delineated and surgically resected, it will help delay the need of chemo- and/or radiotherapy and consequently reduce their potential deleterious effects on the QoL of the patients. Moreover, if glioma imaging becomes more accurate regarding the identification of PTZ and malignant foci, it will help in deciding the correct timing and duration of adjuvant treatment, as tumor progression would be more firmly detected [78]. A different idea concerning chemotherapy entails its use on DLGG involving eloquent areas and/or too diffuse that the initial total or subtotal resection is not feasible [78]. This thinking is based on the fact that chemotherapy may reduce peripheral tumor infiltration on the PTZ and afterwards allows a more significant EOR [79]. In line with this rationale, if imaging techniques detected a very wide PTZ, most likely in predominantly infiltrative tumors, chemotherapy could be considered a neoadjuvant treatment as an attempt to make the tumor less diffuse, thus reopening the door to a subsequent surgery with a better resection. The same logic could be applied before a foreseen second surgery if imaging could detail the pattern of invasion of the tumoral regrowth as largely infiltrative. Interestingly, several series in the literature described the usefulness of neoadjuvant chemotherapy, which, in addition, is able to preserve the cognitive status [80,81,82].

## 6. PTZ and Epilepsy on DLGG

Epileptic seizures are the most common symptom at the time of diagnosis of DLGG, occurring in more than 90% of cases [25,83,84,85,86]. DLGGs are thought to be more epileptogenic than HGG, as slow-growing tumor cells may have intrinsic epileptogenic properties [24].

Glioma-related epileptogenic mechanisms are an important matter of study. They are known to be not only multifactorial, but intermixed and dependent of the characteristics of the tumor itself as well as an alteration of the surrounding brain (structurally and functionally) due to edema, blood–brain barrier disruption, and inflammatory reactions [25,87]. Furthermore, specific pathophysiological changes occurring in the peritumoral area (such as neuronal excitability, disrupted transmitter release, neurotransmitter imbalances, and cystine/glutamate antiporter system [25,29,88]) and epileptogenic foci located within the peritumoral neocortex (in reaction to microscopically infiltrated by ITCs) have been shown to contribute to this pathogenesis [25,86,89,90]. A reason to consider the importance of the interactions between the tumor and the neocortex rather than the intrinsic glioma properties alone in DLGG epileptogenicity is the lack of significant correlations between seizures and tumor volume, edema, mass effect, necrosis, histopathological, or molecular findings, but a positive correlation to tumor location and cortex involvement [25,86,87,89,91].

In the same way, preoperative and intraoperative electrophysiological recordings demonstrate that epileptic seizures arise from the peritumoral neocortex and not from the tumor core [83,89], as reported by Wang et al. (2021) in a patient that underwent cortical electroencephalography monitoring during a craniotomy for DLGG, as well as in the tumor-bearing cortex of 12 model rats [29]. Moreover, ex vivo electrophysiological explorations of spatially oriented human samples also support epileptic activities arising mainly within the peritumoral neocortex infiltrated by glioma cells but not in the glioma core [91,92,93].

Structural reorganization and functional deafferentation with neuronal and glial losses, neurogenesis, reactive astrogliosis, and neuronal, axonal, and synaptic plasticity within the peritumoral neocortex have been described, resulting mainly in a reduction of inhibitory pathways and in an increase of excitatory ones [87,91,94,95]. Furthermore, altered neurotransmitter homeostasis in the PTZ may contribute to tumor-related epileptogenesis [24]. It has been suggested that the tumor cells themselves are neurotoxic (through cytokine and neurotransmitter release) and can elicit seizures by disruption of neuronal pathways or through local glutamate concentration/metabolism changes. A decrease in glutamate uptake and an increase in its release by glioma cells and by neighboring nontumoral astrocytes and activated microglia lead to an excessive glutamatergic excitatory neurotransmission and to a higher risk of seizures [28,91,96,97]. Moreover, GABAergic signaling in the PTZ, affected by the extracellular glutamate, is also involved in both glioma growth and epilepsy [25,89,91,92,97,98,99]. Additionally, alkalization of the peritumoral neocortex [87], synapse-associated proteins [28], and activation of the mammalian target of rapamycin (mTOR) pathway [25] have been implicated in epileptogenesis. Dey et al. (2021) demonstrated increased spontaneous glutamatergic and GABAergic synaptic activity onto pyramidal neurons in the peritumoral samples of DLGG in patients with a history of seizures as compared with those in patients without seizures [27].

It is nowadays accepted that the EOR is the main predictor of postoperative seizure control in DLGG [12,18,86,100]; therefore, its positive impact on QoL should not be overlooked. It has been shown that, in addition to its oncological impact, complete surgical resection is a predictor of epileptic seizure control, as demonstrated in two monocentric studies of 332 and 508 patients, respectively, and in a systematic literature review with meta-analysis, pooling 773 patients from 20 small-sized studies [101,102,103]. In their review of 346 DLGGs, Still et al. (2019) found out that postoperative seizure control is more likely when EOR is ≥91% and/or when the residual tumor volume is ≤19 cm^3^ in supratentorial DLGG gliomas presenting with seizures. However, they did not contemplate resected peritumoral cortex analysis [104].

Considering all the foregoing, SpTR in DLGG has been proposed not only to avoid malignant transformation, but also for better seizure control [12,18,25]. In their cohort of 449 presumed DLGGs, Rossi et al. (2020) observed a significant improvement of epilepsy control in the SpTR group (91.7% vs. 77.5%, *p* < 0.05): a consistent decrease in the number and frequency of seizures was found in the “supratotal” group. In this context, the number of antiepileptic drugs (AEDs) required to obtain seizure control was also lower, indicating that removal of brain parenchyma surrounding the tumor affords better epileptic long-term outcomes, particularly in cases of preoperative long-standing and intractable epilepsy [19]. Indeed, in 16 DLGG consecutive patients who benefited from SpTR, the rate of postoperative seizures was null, with arrest of or at least decrease in AED dosage in all cases [18].

Additionally, in some occasions, the epileptogenic zone may be even farther than the PTZ, including presumed epileptogenic foci beyond the tumor (e.g., hippocampal formation for paralimbic DLGG) [25,105,106]. Ghareeb et al. (2012) demonstrated seizure control in all cases of hippocampectomy of seven patients with intractable epilepsy generated by a paralimbic DLGG, even if the glioma did not seem to invade the hippocampus on the preoperative MRI: all of them resumed their social and professional activities after surgery (while they were not able to work before), in comparison with only 62.5% of the patients who did not undergo hippocampectomy [106]. Even so, in these particular cases, the functional benefit of complete seizure control, allowing patients also to keep their driving license [26,86], has to be weighed against the functional risk of verbal memory decline [86].

Finally, the capacity of oncological adjuvant treatments also needs to be taken into account for seizure control. Conventional radiotherapy has been reported to help epilepsy relief in about 75% of DLGG patients with uncontrolled seizures [84,107]. In addition, chemotherapy with alkylating agents allowed a reduction of the seizure frequency in 50%–60% of patients (20%–40% of them being seizure free), with a better control than patients who are on AEDs only [80,108,109,110,111].

## 7. Conclusions and Future Perspectives

The definition of the PTZ as a peripheral area with macroscopic characteristics of normal brain but harboring microscopic tumor infiltration is an oversimplified description for a much more complex entity with particular cellular and metabolic features, in which glioma cells and neural tissue strongly interact.

Although most studies about PTZ have been conducted on HGG, new significant insights emerged: on the distinctive distribution of immune cells, including immunosuppressive cells, within the PTZ [112]; on the metabolic characteristics of the tumoral tissue and of the PTZ that are being increasingly explored through radiomics and radiogenomics [38,39]; on the gene expression pattern and mutational landscape of the PTZ different from the glioma core and surrounding brain tissue [15]; and also on the understanding of neurogliomal synapses and their role in brain tumor progression, especially the glutamatergic pathways [113,114]. Tumor sample examination, more specifically regarding the relationship between glioma cells and healthy tissue, must continue to evolve in the next years to expand our understanding of the PTZ not only in HGG but also in DLGG, as this knowledge is paramount to help determine new therapeutic targets. In fact, the development of connectome-guided supratotal surgical resection and original drugs specifically targeting the peritumoral tissue, such as immunotherapy taking into account the glioma microenvironment [115] or new AEDs [25], may represent a major treatment advancement that could both improve QoL (including seizure control) and increase survival in (low-grade) glioma patients. To this end, progress in glioma imaging is of utmost importance to improve the demarcation of the PTZ. In the past years, whereas a large amount of radiological explorations mainly tried to differentiate LGG and HGG, new studies aiming at noninvasively identifying ITCs and their spatiotemporal patterns of progression should be encouraged. Furthermore, multimodal explorations combining neuroimaging and genetic analyses must be developed in a more systematic way [15].

In summary, further investigations of the PTZ, a specific entity at the interface between neuro-oncological (tumor cell migration), neurosurgical (intrasurgical mapping), and neuroscientific (brain connectome) considerations, for a long time neglected especially in DLGG, are needed to optimize the onco-functional balance of multimodal therapeutic management in order to maximize the long-term outcome both regarding OS and QoL.

## Figures and Tables

**Figure 1 brainsci-12-00504-f001:**
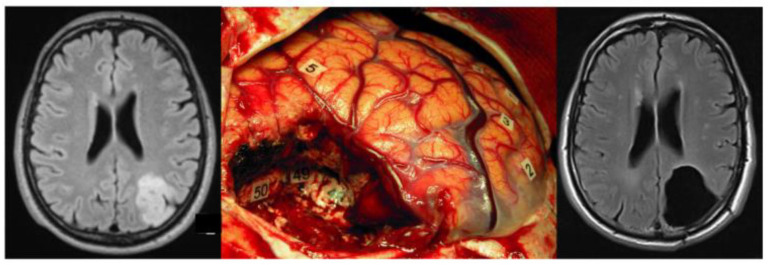
Connectome-based supratotal resection of a left parietal LGG achieved in awake patient. Left: Preoperative axial FLAIR-weighted MRI achieved in a 54-year-old man, engineer, who experienced seizures that allowed the discovery of a left parietal DLGG. The neurological examination was normal. The tumor volume was 35 cm^3^. Middle: Intraoperative view after resection in awake patient. The anterior part of the left hemisphere is on the right, and its posterior part is on the left. The resection was achieved according to functional boundaries, identified using direct electrical stimulation both at the cortical level (5: naming site) and at the subcortical level as follows: Tag 48: somatosensory fibers inducing dysesthesia of the right upper limb when stimulated, and representing the anterior and deep limit of resection; Tag 49: posterior part of the arcuate fasciculus generating phonemic paraphasia when stimulated, and representing the anterolateral and deep limit of resection; Tag 50: optic tracts, eliciting visual disturbances when stimulated, and representing the inferior part of resection. Right: Postoperative axial FLAIR-weighted MRI (performed 3 months after resection) demonstrating a supratotal resection, that is, with removal of a margin all around the preoperative FLAIR abnormality. The patient resumed a normal familial, social, and professional life within 3 months after surgery. A diffuse WHO grade II oligodendroglioma (1p19q codeleted) was diagnosed, and no adjuvant treatment was administrated. The imaging is stable with 13 years of follow-up, and the patient continues to enjoy an active life, with no symptoms.

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
