# Peer review of "The Concept of «Peritumoral Zone» in Diffuse Low-Grade Gliomas: Oncological and Functional Implications for a Connectome-Guided Therapeutic Attitude"

_brainsci, 2022, doi:10.3390/brainsci12040504_

Round 1

Reviewer 1 Report

Author report on the contemporary role of peritumoral zone (PTZ) in Low Grade Diffuse Gliolan (LGDG). They mainly emphasizehow it should be considered the target for LGDG therapy. Surgery aimed to a Supra total resection is the main option to reduce malignant transformation and quality of life. Even chemotherapy and radiotherapy may play a role trough interaction with PTZ more than with tumoral core.

I believe this manuscript should became the major reference for neuroscientists, neurosurgeon and oncologists involved in the management of patients harboring LGDG.  

Author Response

We thank the Reviewer 1 for his/her positive comments.

Reviewer 2 Report

I sincerely congratulate the authors for this interesting Review in this field of research.

I suggest the authors to add, in the 2nd paragraph, a reference regarding the most recent WHO CNS5 tumor classification, discussing its molecular innovation in tumor discrimination.

Author Response

We thank the Reviewer 2 for his/her positive comments. The following reference regarding the most recent WHO CNS tumor classification has been added in the 2nd paragraph: “Louis et al. The 2021 WHO Classification of Tumors of the Central Nervous System: A Summary. Neuro Oncol 2021, 23 (8), 1231–1251”. Molecular innovation in glioma discrimination has been mentioned, by insisting on the fact that “The 2021 fifth edition of the WHO classification introduced major modifications that advance the impact of molecular diagnostics in brain tumor, it established different approaches to glioma nomenclature and grading, and it emphasized the importance of integrated diagnoses: new tumor types and subtypes were introduced, based upon new diagnostic technologies such as DNA methylome profiling.”

Reviewer 3 Report

This review focuses on the available literature related to the peritumoral zone in low grade diffuse glioma.  The content is interesting and leads to conclusions regarding potential future research and treatments.  The focus on low grade glioma is especially significant given how neglected this population is within the previous literature in terms of predicting outcomes.

I would like to see some discussion regarding T1 weighted imaging given that it is standard of care for glioma and was the only sequence not mentioned.  Even if T1 MRI with or without contrast is not particularly useful for detecting PTZ, this is important to note given that there are emerging methods focused on radiomics and connectomics of T1 MRI.

Speaking of connectomics, the references to the connectome within this manuscript are very confusing.  Connectome refers to brain network connectivity and in the context of neurosurgery typically means the avoidance of highly connected nodes or hubs and/or underlying functional connections that are not always evident from localized neuroanatomical boundaries.  It is very unclear what the authors mean by “connectome” within their manuscript.  For example, Figure 1 describes electrical stimulation as “connectome-based” neurosurgery which seems inaccurate as is.  Electrical stimulation results do not necessary infer connectivity.  Several previous studies have employed connectome analyses to demonstrate that molecular variants of glioma are associated with different profiles of brain network disruption and infiltration and even to predict tumor mutation and patient survival.  However, it is unclear how the connectome pertains to the PTZ in the present manuscript. 

There are some typos throughout such as “T2-weigted” and “FET PET” and some of the grammar should be made more formal (e.g. “Talking about MRI”)

Author Response

Comment 1:
This review focuses on the available literature related to the peritumoral zone in low grade diffuse glioma. The content is interesting and
leads to conclusions regarding potential future research and treatments. The focus on low grade glioma is especially significant given how
neglected this population is within the previous literature in terms of predicting outcomes.

Authors’ Response:

We thank the Reviewer 3 for his/her positive comments.

Comment 2:

I would like to see some discussion regarding T1 weighted imaging given that it is standard of care for glioma and was the only sequence
not mentioned. Even if T1 MRI with or without contrast is not particularly useful for detecting PTZ, this is important to note given that
there are emerging methods focused on radiomics and connectomics of T1 MRI.

Authors’ Response:

Despite the fact that T2/FLAIR is the standard of care for low-grade glioma (see Karschnia et al., Evidence-based recommendations on
categories for extent of resection in diffuse glioma. Eur J Cancer, 2021), and that T1 MRI is not helpful for detecting PTZ, it has
nonetheless been mentioned in paragraph 3 that “Of note, there are emerging methods based upon deep learning and radiomic model which
could be promising for glioma grading using multiplanar reconstructed MR contrast-enhanced T1-weighted imaging (Ding et al., Quant
Imaging Med Surg, 2022).”

Comment 3:

Speaking of connectomics, the references to the connectome within this manuscript are very confusing. Connectome refers to brain
network connectivity and in the context of neurosurgery typically means the avoidance of highly connected nodes or hubs and/or
underlying functional connections that are not always evident from localized neuroanatomical boundaries. It is very unclear what the
authors mean by “connectome” within their manuscript. For example, Figure 1 describes electrical stimulation as “connectome-based”
neurosurgery which seems inaccurate as is. Electrical stimulation results do not necessary infer connectivity. Several previous studies
have employed connectome analyses to demonstrate that molecular variants of glioma are associated with different profiles of brain
network disruption and infiltration and even to predict tumor mutation and patient survival. However, it is unclear how the connectome
pertains to the PTZ in the present manuscript.

Authors’ Response:
It has extensively been demonstrated that intraoperative direct electrical stimulation (DES) represents a unique opportunity to directly
investigate cerebral functional connectivity (Duffau, Stimulation mapping of white matter tracts to study brain functional connectivity.
Nature Reviews Neurology, 2015), resulting in the proposal of new models of brain connectome based upon the dyssynchronization within
and between neural networks elicited by DES (Herbet and Duffau, Revisiting the Functional Anatomy of the Human Brain: Toward a
Meta-Networking Theory of Cerebral Functions. Physiological Reviews, 2020). In clinical practice, this novel concept led to a
paradigmatic shift, that is, to achieve a surgical resection up to the identification of the neural circuits critical for brain processing identified
by DES (Sarubbo et al. Mapping critical cortical hubs and white matter pathways by direct electrical stimulation: an original functional
atlas of the human brain. Neuroimage, 2020), and not according to the data provided by neuroimaging, which underestimates glioma
infiltration: this is the principle of “connectome-based surgery”, enabling to achieve supratotal resection (and not “DLGG core-based
surgery”) if functionally feasible, namely, with removal of the PTZ in addition to the tumor visible on preoperative MRI (Duffau, Brain
connectomics applied to oncological neuroscience: from a traditional surgical strategy focusing on glioma topography to a meta-network
approach. Acta Neurochir (Wien), 2021).

This critical issue has been added in paragraph 4.

Comment 4:

There are some typos throughout such as “T2-weigted” and “FET PET” and some of the grammar should be made more formal (e.g.
“Talking about MRI”)

Authors’ Response:

These typos have been corrected.

Reviewer 4 Report

This review describes the current state-of art regarding the peritumoral zone in diffuse low-grade gliomas. The authors investigated the oncological and functional implications of this issue in multimodal therapeutic management. This argument has been covered in a comprehensive way. Evident efforts have been made to cover this argument considering the extensive literature in this field. The paper is globally interesting and fluently written. I don't have particular suggestions for the authors since I found the article a good scientific read globally.

Author Response

We thank the Reviewer 4 for his/her positive comments.

Round 2

Reviewer 3 Report

The authors have adequately addressed my previous concerns.